# Unlocking Neural Function with 3D In Vitro Models: A Technical Review of Self-Assembled, Guided, and Bioprinted Brain Organoids and Their Applications in the Study of Neurodevelopmental and Neurodegenerative Disorders

**DOI:** 10.3390/ijms241310762

**Published:** 2023-06-28

**Authors:** Chiara D’Antoni, Lorenza Mautone, Caterina Sanchini, Lucrezia Tondo, Greta Grassmann, Gianluca Cidonio, Paola Bezzi, Federica Cordella, Silvia Di Angelantonio

**Affiliations:** 1Department of Physiology and Pharmacology, Sapienza University of Rome, 00185 Rome, Italy; chiara.dantoni@uniroma1.it (C.D.); lorenza.mautone@uniroma1.it (L.M.); tondo.1855708@studenti.uniroma1.it (L.T.); paola.bezzi@uniroma1.it (P.B.); 2Center for Life Nano- and Neuro-Science of Istituto Italiano di Tecnologia (IIT), 00161 Rome, Italy; caterina.sanchini@iit.it (C.S.); greta.grassmann@iit.it (G.G.); gianluca.cidonio@iit.it (G.C.); 3Department of Biochemical Sciences “Alessandro Rossi Fanelli”, Sapienza University of Rome, 00185 Rome, Italy; 4Department of Fundamental Neurosciences, University of Lausanne, 1011 Lausanne, Switzerland; 5D-Tails s.r.l., 00165 Rome, Italy

**Keywords:** neuronal development, cortical organoids, iPSC, neurons, astrocytes, microglia, Alzheimer’s disease, 22q11 syndrome, bioprinting

## Abstract

Understanding the complexities of the human brain and its associated disorders poses a significant challenge in neuroscience. Traditional research methods have limitations in replicating its intricacies, necessitating the development of in vitro models that can simulate its structure and function. Three-dimensional in vitro models, including organoids, cerebral organoids, bioprinted brain models, and functionalized brain organoids, offer promising platforms for studying human brain development, physiology, and disease. These models accurately replicate key aspects of human brain anatomy, gene expression, and cellular behavior, enabling drug discovery and toxicology studies while providing insights into human-specific phenomena not easily studied in animal models. The use of human-induced pluripotent stem cells has revolutionized the generation of 3D brain structures, with various techniques developed to generate specific brain regions. These advancements facilitate the study of brain structure development and function, overcoming previous limitations due to the scarcity of human brain samples. This technical review provides an overview of current 3D in vitro models of the human cortex, their development, characterization, and limitations, and explores the state of the art and future directions in the field, with a specific focus on their applications in studying neurodevelopmental and neurodegenerative disorders.

## 1. Introduction

The human brain is one of the most complex organs in the body and is the center of the central nervous system. Understanding its function and disease processes, especially neurodevelopmental and neurodegenerative disorders, is a major challenge in modern neuroscience. While traditional methods such as animal models and post-mortem studies have contributed greatly to our knowledge, they have limitations and may not fully reflect the human condition. As a result, there is a growing need for in vitro models that can simulate the complex structure and function of the human brain.

Three-dimensional (3D) in vitro models of the human brain, such as organoids, cerebral organoids, three-dimensional bioprinted brain models, and functionalized brain organoids, have the potential to provide a platform for the study of human brain development, physiology, and disease. These models can recapitulate various aspects of human brain anatomy, gene expression, and cellular behavior, making them useful tools for drug discovery and toxicology studies [1]. Moreover, 3D in vitro models allow for the exploration of human-specific phenomena that cannot be studied in animal models. The use of 3D in vitro models for the study of neurodevelopmental and neurodegenerative disorders has gained significant attention in recent years thanks to the availability of different types of stem cells.

Human pluripotent stem cells (hPSCs), which include embryonic stem cells (ESCs) and induced pluripotent stem cells (iPSCs), are known for their remarkable self-organization and ability to form neural cells as a default pathway when spontaneously aggregated into 3D spheres [2]. Yoshiki Sasai’s lab made a breakthrough by demonstrating how ESCs cultured in 3D aggregates could spontaneously generate highly organized brain structures [3,4]. Furthermore, by utilizing the extracellular matrix support of Matrigel, Lancaster and colleagues created a more complex 3D tissue from hPSCs, called a cerebral organoid, that closely resembles a growing human brain [5]. These developments in stem cell research and our understanding of developmental biology have led to the creation of various techniques for generating brain regions over the past decade. These techniques have been extensively reviewed [6,7,8,9,10,11] and are generally categorized into two groups based on patterning approaches. Unguided protocols rely on the intrinsic differentiation of PSCs without growth factor patterning, resulting in brain organoids that contain a diverse mixture of neurons, glia, photoreceptor cells, and even cells of non-ectodermal origin [5]. However, this method is hampered by variability in the emergence and size of different brain regions, making it difficult to study the development and function of specific brain structures. On the other hand, guided protocols use small molecules and patterning growth factors to direct cells towards a specific fate, resulting in the enrichment of a region of interest [4,12]. A wide range of region-specific brain organoids have been generated using guided protocols, including the cortex [12,13], cerebellum [14], hippocampus [15], pituitary gland [16], hypothalamus [17,18], spinal cord [19], thalamus [20,21], choroid plexus [22], striatum [23], and optic cup [24]. The choice of protocol for generating cerebral or region-specific brain organoids depends on the scientific questions being addressed. Nonetheless, the ability to generate different brain regions with distinct cellular identities from PSCs in vitro provides access to human-specific brain development and pathophysiology, which was previously limited by the scarcity of embryonic human brain samples.

In this technical review, we aim to provide an overview of current 3D in vitro models of the human cortex, including organoids, cerebral organoids, 3D bioprinted brain models, and functionalized brain organoids, and their applications in the study of neurodevelopmental and neurodegenerative disorders. The review will cover various aspects of these models, including their development, characterization, and limitations, and will discuss the current state of the art and future directions in this rapidly evolving field.

## 2. From 2D to 3D: The Rise of Cerebral Organoids for Studying Human Brain Development

The development and maturation of the human brain are complex processes that rely on the intricate interactions between neuronal and glial cells and the establishment of connections with other brain regions. Two-dimensional (2D) stem-cell-based models have contributed to our understanding of different molecular processes involved in human brain development, function, and neurodegeneration [25,26,27,28,29], but these models lack the complex cytoarchitecture of the cerebral cortex, limiting their ability to study the 3D interaction of brain cells during development. To address this limitation, human-derived brain organoids, which are self-assembled, organized structures composed of neuronal and glial cells, have emerged as valuable tools to study the developing brain under both physiological and pathological conditions, enabling the study of previously inaccessible aspects in a controlled laboratory setting (for a systematic review, see [30]).

Since the early 1990s, researchers have been actively pursuing methodologies to induce neural differentiation from pluripotent stem cells (PSCs) in three-dimensional (3D) environments [31]. However, it is important to note that not all 3D neural culture systems can be classified as brain organoids. Neurospheres, for instance, are 3D aggregations of various cell types derived from neural progenitor cells within the central nervous system (CNS) [31]. The neurosphere culture system has been extensively utilized for comprehensive investigations into the proliferation, self-renewal capabilities, and differentiation potential of neural stem cells [32,33]. Neurospheres represent an in vitro model system consisting of clusters of neural progenitor cells (NPCs), including neural stem cells [34]. These neurospheres are derived from isolated primary tissue or low-passaged NPCs obtained from induced pluripotent stem cells (iPSCs) and are cultured for limited periods before being utilized in various research applications. Neurospheres have been widely employed as the conventional method for NPC culture, with over 2000 publications since the original report by Reynolds [31]. They are particularly favored for obtaining NPCs from neurogenic areas in animal and human tissues [35]. Typically cultured in serum-free medium supplemented with fibroblast growth factor 2 (FGF-2) and epidermal growth factor (EGF), neurospheres do not require an adherent substrate that supports NPC expansion. This unique culture system provides an unparalleled platform to assess the stem-cell-like behavior of neurogenic tissue and enables investigations into the molecular and cellular characteristics of NPCs for in vivo transplantation in mice and humans [36,37,38]. However, despite their widespread use, maintaining neurosphere cultures can be challenging due to their rapid growth and substantial apoptosis. NPC cultures necessitate frequent dissociation and passaging every 7–10 days to control the neurosphere size and prevent excessive cell death. It is important to note that neurosphere cultures are most useful within the first 5–10 passages, as prolonged passaging can lead to aneuploidy and the selection of clones that may alter the intrinsic properties of NPCs [35]. Furthermore, neurospheres are limited in their ability to determine the self-renewal capacity of NPCs due to the fusion of neurospheres in vitro, making them non-clonal [34]. Criticism has been directed towards the use of neurospheres in vitro, suggesting that neurosphere formation may be an artifact of any cultured cell in the absence of a substrate and under the influence of selected growth factors, thus lacking intrinsic biological significance.

However, the emergence of cerebral organoid technology as a model system for human brain development provides new evidence to support the contention that neurosphere 3D aggregates may indeed convey biological relevance (see Table 1). Overall, while neurospheres have been widely utilized and serve as a valuable tool for NPC culture and exploration, their maintenance can be challenging, and there are limitations regarding their clonality and biological significance. The advent of cerebral organoids offers an exciting avenue for further understanding the biological implications of neurosphere 3D aggregates.

Recent advances have led to the development of 3D in vitro models that recapitulate the hallmarks of the developing cerebral cortex [39] (Figure 1). Human pluripotent stem cells (hPSCs) have an inherent tendency to differentiate into the neuroectodermal lineage and generate neural tissue even in the absence of external patterning factors, thanks to their “neural default pathway” [9,40]. Lancaster and Knoblich [40] described the development of cerebral organoids from hPSCs by utilizing the self-organization and self-patterning ability of hPSCs, along with minimalistic media and matrix-embedding techniques. Their protocol led to the generation of neuroectodermal tissue, primarily in the form of neural rosettes, which recapitulated the tissue architecture of the germinal zones of neural stem and progenitor cells, with differentiated neurons migrating outward. This technique, which produced organoids with broad regional identities, was thus named cerebral organoids. These 3D structures, thanks to Matrigel droplet embedding, are able to perform apicobasal expansion of neuroepithelial buds through the basement scaffold of the extracellular membrane provided by Matrigel. After the subsequent transfer to a rotating bioreactor for greater nutritional absorption, the neuroepithelial regions began to generate fluid-filled cavities similar to ventricles. Over time, these structures became increasingly complex generating populations of cortical progenitors. For terminal differentiation, retinoic acid was added, and several populations of neural progenitors appeared, including radial glia cells, the progenitor cells responsible for the formation and the correct alignment of cortical neurons in space. At this point, in some regions of the organoid, the ventricular (VZ) and subventricular zones (SVZ) were formed and, similar to neurogenesis in vivo, the neural progenitors began to migrate and differentiate by forming the layers of the cortical plate [8]. As well as the neuronal layers of the cerebral cortex and because of the lack of external inductive signals, a variety of regional identities are also present in whole-brain organoids. These structures spatially and functionally resemble different regions of the developing brain including the hindbrain, midbrain, forebrain, and retinal tissues.

Interestingly, the different brain identities present inside the cerebral organoids are not randomly dispersed; in fact, some of the neighboring regions have boundaries that resemble borders found in vivo. During brain development, the presence of multiple organizing centers near the neuroepithelial sheet has a crucial role in releasing different morphogen gradients responsible for brain patterning. The different combinations of several signaling factors will affect the cellular-specific regional identities. Among the organizing centers, hem and antihem are responsible for telencephalic signaling. The former is located at the midline adjacent to the choroid plexus and dorsal telencephalon and releases bone morphogenetic proteins (BMP) and Wnts involved in dorsal identities. The latter instead sits opposite the hem and separates dorsal and ventral telencephalic regions through the expression of various morphogens including Wnt antagonists. In cerebral organoids, the VZ-like structure has abrupt borders between the dorsal (TBR2+) and ventral (GSX2+) forebrain identities as would be found at the antihem, as well as being tissue-positive for Wnt2b and BMP6, molecules produced by the hem in vivo, observed adjacent to choroid plexus, which was immediately followed by the presence of dorsal telencephalic tissue (TBR2+) [41]. These suggest that self-patterned organoids are able to develop into complex brain architectures without any cues or a body axis for reference. Neuroepithelial tissue is also capable of spontaneously setting up signaling centers and developing local tissue patterning.

A major limitation of cerebral organoids is their high batch-to-batch variability due to the stochastic nature of the spontaneous differentiation of human pluripotent stem cells (hPSCs) [42]. This unpredictability and inconsistency in development can be seen even when comparing brain organoids from the same differentiation batch, affecting the model’s repeatability and applicability [5,43,44]. To mitigate this issue, researchers have developed various methods to decrease or eliminate batch-to-batch variability in cerebral organoids. For example, Lancaster and colleagues successfully improved and standardized the workflow of cerebral organoid generation using poly(lactide-co-glycolide) copolymer fiber microfilaments to generate elongated embryoid bodies (EBs) [45]. They discovered that organoids at the embryoid body stage were relatively homogeneous, whereas variability, especially between batches, increased during neural induction. This increase in variability may be due to the low surface-area-to-volume ratio, which can influence neuroectoderm development on the exterior of the embryoid body.

Taking advantage of recent findings on micropatterned substrates, Lancaster and colleagues engineered microfilament-based cerebral organoids (enCORs) with increased surface area, dense cell composition, and polarized neural ectoderm formation. Notably, the efficiency of neuroectoderm formation was improved, and the amounts of endoderm and mesoderm identities decreased, in contrast to whole-brain organoids, which typically develop non-ectodermal identities. The quantification of these identities revealed the reproducible production of neuroectoderm in enCORs with minimal non-neural tissues, whereas spheroids had highly variable levels of all germ layer identities [45].

Another important limitation of cerebral organoids is the lack of a homogeneous and deep oxygen and nutrient supply, leading to the necrosis of cells deep within the organoid. To overcome this limitation, Lancaster and colleagues developed a novel method based on the organotypic slice culture technique, generating air–liquid interface cerebral organoids (ALI-COs). They demonstrate that ALI-COs improve neuronal survival and morphology compared to whole brain organoids when kept in culture for several years. In fact, they notice that whole-brain organoids after months of growth show a significant loss of neurons and start to accumulate reactive astrocytes on the edge of the organoid. In contrast, ALI-COs show an extensive axon outgrowth reminiscent of in vivo nerve tracts. Additionally, astrocytes grown inside ALI-COs displayed healthier morphology, with numerous fine processes. These findings suggest that ALI-COs could be a useful in vitro model for investigating the later stages of neuronal maturation and neurological diseases [44,46].

## 3. Beyond Batch Syndrome: Guiding Cellular Fate in Brain Organoid Development

Self-assembled organoids were found to be limited in their usefulness for modeling specific brain regions and therefore for studying the regional mechanisms of brain development, due to their lack of spatial organization. To overcome these limitations, researchers have developed novel 3D brain models modulating specific morphogen-related signaling to commit iPSCs towards a specific neuronal fate. These organoids can be directed towards the development and maturation of specific brain regions and are named patterned organoids [12,17] (Figure 1).

To produce patterned organoids, single pluripotent stem cells, neuronal progenitors, or neuroepithelial stem cells are seeded in microwells, such as Aggrewell, U-bottom wells, or 3D-printed supports, to form embryoid bodies (EBs). The concurrent use of inhibitors of the BMP, TGF-ß, and WNT signaling pathways, respectively, accelerates the induction of the differentiation into forebrain lineage cells [21,47]. Indeed, SB-431542, LDN-193189, and XAV939 prompt cells to activate a differentiation program that directs their development and maturation into neurons or glial cells of the cerebral cortex; additionally, the addition of SHH and CHIR-99021 commits cells to differentiate in midbrain cells, thereby avoiding the “batch syndrome” that characterizes the generation of unguided brain organoids. Specifically, by inhibiting TGF-β signaling using SB-431542, BMP signaling with LDN-193189, and Wnt signaling with XAV939, it is possible to promote the generation of specific cortical neuronal populations, regulating the balance between stem cell self-renewal and differentiation and thus the formation of cortical layers within the organoids. Pasca and colleagues in 2015 generated dorsal forebrain-like structures from pluripotent stem cells, called human-cortical spheroids (hCSs) via the addition of small molecules such as dorsomorphin and SB431542, which inhibit bone morphogenic proteins (BMPs) and TGF-β signaling, enhancing the neuroectoderm fate. It has been demonstrated through the analysis of hCSs’ transcriptional profiles that they exhibit developmental maturity and regional identity during their maturation into two distinct time points such as day 52 and day 76 of hCS development. In particular, they found a strong overlap between hCSs and cortical developmental stages up to late–mid-fetal periods (19–24 PCW). Qian and colleagues performed a large-scale comparison of transcriptome datasets between organoids and 16 different human brain areas at different developmental stages, revealing a temporal correlation between hCSs and fetal human brain development, particularly in the prefrontal cortex. In this regard, organoids at day 50 showed a transcriptional profile closely related to the prefrontal cortex at post-conception week (PCW) 8–9, while organoids at day 100 were more similar to brain regions during the 35th PCW. During hCSs maturation, there is an up-regulation of synaptic transmission genes and a down-regulation of genes involved in the cell cycle and cell division [12,17]. Similarly to in vivo corticogenesis, day 14 forebrain organoids show both proliferative zones containing neural progenitors organized inside VZ-like structures as well as GFAP (glial fibrillary acidic protein)-positive extensions resembling radial glia [48]. Moreover, it has been shown that in this time window, hCSs also express forebrain-specific progenitor markers, including PAX6, OTX2, and FOXG1, with minimal expression of markers for other brain regions, while at day 28, there is a consistent increase in TUJ1/CTIP2-positive cells [17]. With further development of the hCSs, the separation of early-born CTIP2 (which labels neurons of layer V)-positive neurons and late-born SATB2+ (which labels neurons of layers II–IV) neurons becomes evident, indicating the specification of deep and upper cortical layers. Immunostaining analysis performed at days 56 and 70 demonstrated that SVZ-containing neurons express low amounts of CTIP2, a transcription factor which is related to migrating neurons, while at day 84, late-born SATB2+ neurons formed a layer partially separated from the early-born CTIP2+ layer, suggesting the specification of upper and deep cortical layers [17].

It has also been demonstrated that glial cells, including astrocytes and oligodendrocytes, are present within cortical organoids. Astrocytes have been recently demonstrated to be crucial in the regulation and modulation of synaptogenesis [49,50]. Astrogenesis, as well as the maturation of astrocytes, is indeed another key point for proper corticogenesis [51,52]. Several studies revealed the presence of astroglial cells, positive for GFAP and S100 calcium-binding protein-β (S100b) [12,53], thus suggesting that cerebral organoids are valuable tools for studying astrogenesis and the role of astrocytes in corticogenesis. Finally, electrophysiological recordings on both whole and sliced organoids at day 130 have shown that cells exhibit spontaneous synaptic activity and are capable of generating repetitive action potentials when depolarized, confirming the presence of a proper neuronal network and synaptic functionality [54].

## 4. Improving Brain Organoids by Introducing Microglia and Vascularization

Brain organoids face a significant drawback due to their absence of vascularization and microglia, which are crucial components in the development and functioning of the brain. Microglia are essential for maintaining homeostasis and the immune response in the brain [55,56,57,58,59,60], and the absence of microglia in brain organoids limits not only their ability to accurately model all brain diseases characterized by the presence of a neuroinflammatory state but also the correct brain development and function in physiological conditions.

Microglia, the primary neuroimmune cells in the brain, originate from erythro-myeloid progenitors in the embryonic yolk sac and differentiate into microglia after migrating into the developing brain. Microglia are sustained through self-renewal processes dependent on cytokines like IL-34 and CSF-1, as well as transcription factors such as PU.1 and interferon-regulatory factor 8. Their primary role is immune surveillance, where resting microglia constantly monitor the brain environment, while activated microglia respond to pathological insults and eliminate harmful species through various mechanisms of phagocytosis. Furthermore, microglia contribute to neural development, synaptic formation and plasticity, and neural network maturation. They regulate neural development by colonizing cortical proliferative zones and phagocytosing neural precursor cells. During postnatal brain development, microglia actively prune weak synapses, thereby shaping neuronal circuits [58,60,61].

Neuronal lineage cells originate from the ectoderm, while differentiation into endoderm and mesoderm lineages is typically suppressed in the formation of human brain organoids. Consequently, microglia, which arise from non-neuroectodermal origins, are usually absent in brain organoids. However, recent advancements have led to the development of strategies to generate microglia-containing brain organoids, offering valuable insights into microglial functions and their relevance to brain disorders. Several strategies have been developed to address this limitation, such as incorporating microglia into organoids through co-culture, genetic modification, or separate culture insertion. One study conducted by Ormel et al. in 2018 reports the possibility to generate human iPSC-derived brain organoids containing innately developed microglia (oMG), using a protocol that, instead of adding extra BDNF to the differentiation cocktail, like Quadrato et al., reduced the levels of the neuroectoderm stimulant heparin and delayed Matrigel embedment of the organoids [10,62]. This approach resulted in the development of organoids with microglia cells characterized by ramified morphology, microglia-specific marker expression, and a more realistic immune response to inflammation. Another study conducted by Abud et al. in 2017 introduced microglia into cerebral organoids derived from iPSCs through a co-culture system. The resulting organoids exhibited a more mature and functional microglial phenotype validated using a range of methods. Firstly, flow cytometry analysis was employed to assess their phenotype. In addition, the secretion of cytokines/chemokines, specifically IL-1b and IFNγ, was examined subsequent to stimulation with lipopolysaccharide (LPS). Furthermore, to investigate the synaptic pruning capabilities, synaptosome phagocytosis assays were established. These findings provide compelling evidence that co-culture strategies hold significant potential for recapitulating the essential factors involved in the development of tissue-resident microglia in the brain [63]. In 2019, Song et al. used a guided protocol of 30 days (utilizing a differentiation medium containing BMP-4, Activin-A, SCF, and VEGF) to obtain microglial cells derived from iPSCs, which were then introduced into cerebral organoids. To assess the functional activity of these microglia-like cells, the researchers examined their phagocytosis capability, increased expression of TNF-α, and secretion of cytokines in response to pro-inflammatory Aβ42 oligomers, which are known to modulate microglia responses through TREM2 binding. The dorsal or ventral organoids displayed synaptic activities and action potentials, whereas the microglia-like cells exhibited differential migration ability and immune response, with higher TNF-α expression observed in ventral-MG co-cultures. This co-cultured strategy elucidated the impact of microglia on brain organoids, revealing their ability to stimulate cell proliferation and effectively reduce ROS expression, thus closely resembling the microenvironment found in specific brain tissues [64].

An innovative approach was developed by Ao et al. in 2021, where a tubular organoid device was integrated with isogenic microglia to enable hypoxia-free brain organoid culture with continuous medium and oxygen perfusion. The protocol commences by carefully loading human embryonic stem cells (hESCs) into the designated basket within a tubular device. Neural fate induction is accomplished by administering dual-SMAD inhibitors dorsomorphin and A83. In order to direct cellular differentiation towards a forebrain identity, the neural induction medium is subsequently replaced with a composition containing CHIR-99021 and SB-431542. To establish inner lumen fluid flow, the loaded device is placed onto a rocking platform, supplemented with N2 and B27 components. Neuronal characterization involved the assessment of marker expression, specifically, the dorsal forebrain neural progenitor cell markers PAX6 and SOX2, as well as the deep cortical neuron marker TBR1. In contrast, for the characterization of induced microglia, a double staining approach employing Iba1 and CD68 was conducted. Upon LPS/ATP treatment, activated induced microglia also exhibited elevated secretion levels of IL-1β and IL-18, indicating functional inflammasome activation. Additionally, a noteworthy increase in TNF-α secretion was also observed. The present device effectively emulated microglial responses in an environment that closely replicates the inherent microenvironment of the brain. In fact, it was observed that tubular organoids demonstrate a substantial decrease in hypoxia compared to conventional organoids. This finding is particularly intriguing considering that hypoxia is a recognized challenge commonly associated with organoid culture [65].

Until now, a significant limitation in the integration of microglia cells into organoids has been their notable heterogeneity, which impairs reproducibility due to the acquisition of highly divergent cell populations. However, in 2022, Cakir et al. achieved a substantial advancement by inducing the expression of PU.1 in cells, thereby generating microglia-like cells that could be successfully introduced into human cortical organoids. This pioneering approach represents noteworthy progress in addressing the challenge of heterogeneity, ultimately enhancing the reproducibility and consistency of microglia integration within organoid models [66] (Figure 2).

Another weakness of conventional brain organoids is the lack of vascularization, which limits growth, functionality, and immune response. Indeed, without the presence of blood vessels, organoids have a limited capacity for growth and functionality. Indeed, the lack of oxygen and nutrients due to the absence of blood vessels can limit the lifespan of organoids, leading to cell death over time and making it difficult to use organoids for long-term studies [40].

Innovative approaches have successfully developed vascularized organoids capable of performing more complex functions, revolutionizing the field of organoid research. By integrating functional blood vessels within the organoid structures, these advancements have facilitated improved nutrient delivery, waste removal, and oxygenation, closely mimicking the physiological conditions found in real tissues. This vascularization has enabled enhanced cellular interactions, organoid maturation, and the emergence of more intricate multicellular systems, thereby expanding the potential applications and utility of organoids in various biomedical research areas, such as drug discovery, disease modeling, and regenerative medicine. To overcome this limitation, a number of different solutions have been implemented, such as co-culture strategies, 3D-printed microfluidic chips, genome editing, and the fusion of blood vessel organoids with brain organoids. In this section, we will examine the current understanding of the lack of vascularization in organoids, its consequences, and the current progress towards creating more vascularized organoid models.

Pioneering work in the field involved the generation of vascularized organoids by introducing iPSC-derived endothelial cells. This was accomplished through a three-stage process utilizing FGF2, CHIR99012, BMP4, and VEGF to promote proper endothelial differentiation [67]. More recent studies have proposed alternative methods to generate vascularized organoids. One approach involves co-culturing hESCs or iPSCs with umbilical vein endothelial cells, resulting in a high success rate of vascularization exceeding 95%. The vascularized organoids (vOrganoids) exhibit mesh-like and tube-like structures formed by HUVECs, which are identified using markers such as laminin and isolectin I-B4 (IB4) that specifically label blood vessels and endothelial cells [68]. Another effective method involves the integration of differentiated vessels into brain organoids. Ahn et al. developed separate brain organoids and blood vessel organoids and subsequently co-cultured the obtained blood vessel cells with cortical organoids. These vascularized human cortical organoids demonstrate the presence of a functional vascular network, which is absent in regular hCOs [69]. In an alternative approach for generating brain-specific vascular organoids, a guided protocol was utilized to differentiate H9 hESCs into human blood vessel organoids. These blood vessel organoids were then fused with cerebral organoids to create brain-specific vascular organoids. Notably, this technique successfully formed a tightly sealed blood–brain barrier (BBB), consisting of brain microvascular endothelial cells. This closely mimics the BBB function in the selective regulation of the transport of substances to and from the brain and in the protection against harmful agents. The expression of key tight junction proteins, including Claudin5 (CLDN5) and ZO-1, as well as the efflux transporter p-glycoprotein, was examined to evaluate the BBB-like characteristics. These proteins facilitate the recycling of small lipophilic molecules that diffuse into endothelial cells back into the bloodstream [70]. Several additional approaches have been proposed to enhance vascularization in organoids. One strategy involves utilizing 3D-printed microfluidic chips where pericytes and endothelial cells derived from hPSCs spontaneously form well-structured vascular networks. These vascular cells establish physical interactions with cerebral organoids, resulting in the formation of integrated neurovascular organoids on the chip [71]. Another approach involves reprogramming human dermal fibroblasts into endothelial cells using the transcription factor human ETS variant 2 (hETV2). Bilal Cakir et al. (2022) introduced engineered hESCs expressing hETV2 into hCOs, leading to the development of vascularized hCOs with a functional, perfusable vascular-like network. These vascularized organoids exhibit BBB characteristics, such as increased expression of tight junctions, nutrient transporters, and transendothelial electrical resistance [66,72].

Although there are still limitations in the current state of 3D organoids for brain research, the ongoing efforts to improve vascularization hold great promise for the field. These advancements in vascularized organoid development provide opportunities for gaining deeper insights into brain function [68,69,70]. However, it is important to note that in addition to the lack of vascularization, another limitation of organoids is the absence of microglia, which hinders their ability to fully mimic the brain’s cellular composition and interactions [40] (Figure 2).

## 5. Assembling Neural Tissue: Hydrogels and Techniques Used in 3D Bioprinting and Organ-on-a-Chip Technologies

To date, brain organoid techniques still present many challenges: therefore, the convergence between organoid technology and 3D bioprinting could pave the way for optimizing functional 3D in vitro brain models.

Bioprinting is an automated, layer-by-layer deposition of cells embedded in biocompatible materials or bio-inks to fabricate 3D constructs. The development of this technology can also allow for the repair or regeneration of tissues in patients by performing pharmacokinetic studies in vitro [73]. Three-dimensional bioprinting techniques allow for the generation of organized structures, which increase reproducibility. Numerous studies have utilized 3D bioprinting technology to print stem cells, as demonstrated by Reid et al. [74], Gu et al. [75], Nguyen et al. [76], De La Vega [77], and Koch et al. [78]. This approach offers numerous advantages, as embedding ESC and hPSC in a 3D construct maintains their multilineage potential, enabling differentiation and maturation directly within the scaffold. Alternatively, another strategy involves differentiating cells from hPSCs into specific cell types before printing, as described by Faulkner-Jones et al. [79], Ma et al. [80], Yu et al. [81], Ong et al. [82], Moldovan et al. [83], Sorkio et al. [84], Joung et al. [85], and De la Vega et al. [86].

The precise assembly and guidance offered by 3D bioprinting enable the generation of organized structures, improving reproducibility. However, there are significant challenges to overcome, including controlling the relative distribution of cells and the impact of printing on cell viability. In particular, one of the most significant challenges in the bio-fabrication of 3D neural structures is developing bio-ink formulations that enable neuronal survival, differentiation, and maturation while modeling physiological extracellular matrix components and maintaining suitable mechanical properties for printing. In the context of 3D bioprinting, hydrogels are a popular choice for scaffolds due to their ability to retain large amounts of water, biocompatibility, and ability to form networks via the crosslinking of hydrophilic chains. These properties allow for the culturing of neural cells in 3D, which requires a material that can support cell bodies while allowing for process extension and connection. Hydrogels that have been used to print neural cells include alginate, agarose, chitosan ([75,87]), gellan gum-RGD [88], collagen [89], modified gelatin GelMa [90], and Matrigel [85,91] (see Table 2). Matrigel is a solubilized basement membrane matrix that contains many extracellular matrix (ECM) components and is secreted by Engelbreth-Holm-Swarm mouse sarcoma cells. However, this material’s batch-to-batch variability can impact the reproducibility of printing.

There are three main printing methods: laser-assisted, inkjet, and extrusion-based techniques (Figure 3). Laser-assisted bioprinting utilizes two co-planar slides, where a laser is used to transfer bio-ink droplets from the donor to the collector slide. This process is mediated by the formation and expansion of microbubbles on the surface of the donor slide. This nozzle-free technique allows for greater freedom in the choice of bio-ink and results in lower damage to the cells during printing. However, it is also associated with high costs, and slide preparation is complex and time-consuming. The inkjet technique utilizes a drop-on-demand method where micrometric droplets of bio-ink are ejected from the dispenser tip by applying a thermal or mechanical force [93,94,95]. Inkjet was one of the earliest approaches to bioprinting and enables the precise placement of cells with droplets in the picoliter range. However, this technique is only suitable for low-viscosity materials, despite being low-cost and straightforward [17,87]. Extrusion-based bioprinting involves physically extruding the material through the dispenser tip using pneumatic or piston-driven actuators in a continuous manner [96]. This technique is one of the most widely used, but it allows for lower resolution and is particularly suitable for viscous gels. After deposition, materials require a sol-gel transition, from liquid- to solid-like behavior, which can be achieved via direct printing into a cross-linking solution, such as calcium chloride for alginate [97], or using temperature changes or light for photo-cross-linked materials [98]. Coaxial needle printing, which simultaneously extrudes bio-ink and coagulation solution, is an approach that uses ionic crosslinking [99,100].

Hsieh et al. [101] demonstrated that murine NSCs embedded in thermoresponsive water-based polyurethane dispersions could be printed and implanted into a zebrafish brain injury model, leading to the successful rescue of functions of the impaired CNS. Joung et al. [85] combined extrusion-based bioprinting with 3D-printed scaffolds to model the spinal cord. They used an alginate and methylcellulose blend scaffold, and simultaneously printed neuronal progenitor cell (NPC) or oligodendrocyte progenitor cell (OPC) bio-ink into the scaffold, resulting in the development of functional neurons with extensive axon propagation. Gu et al. [75] used an extrusion-based technique to print neuronal constructs. They printed human iPSCs with a bio-ink composed of alginate, chitosan, and agarose, cross-linked in calcium chloride, which allowed in situ proliferation and differentiation into neuroglia and neurons. These cells displayed functional features 30–40 days post-printing. Recent implementations of extrusion 3D bioprinting techniques include their integration with microfluidic devices [102]. Microfluidic-based devices are made of microchannels containing microliter to picoliter volumes of fluids that interconnect chambers where different types of cells can be placed, closely modeling compartmentalized microenvironments. Microfluidic chips have also been demonstrated to be optimal platforms to model endothelial layers and therefore vascularized systems [103,104]. Yu et al. [81] demonstrated that functional tissue structures can be directly printed on microfluidic devices. The integration of 3D bioprinters with microfluidic-based heads allows the precise control of bio-ink volume, the printing of higher cell concentrations [105], the simultaneous extrusion of different materials [100], and the possibility to mimic complex or graded patterns and tissue compositions [106]. De la Vega et al. [77] printed hiPSC-derived NPCs into cylindrical constructs with an extrusion-based microfluidic printer using a fibrin, chitosan, and alginate bio-ink containing purmorphamine- and retinoic acid-loaded microspheres. This allowed for the continuous release of these drugs, leading to differentiation and maturation into spinal motor neurons.

iPSC-derived cortical neurons and glial cells were successfully printed by Salaris et al. in 2019 [91] using a custom extrusion-based bioprinter implemented with co-axial wet-spinning microfluidic devices to print Matrigel/alginate-embedded cells and crosslinking solution simultaneously. The cells could further differentiate within the construct, expressing both neuronal and astrocytic markers, with good long-term cell survival (up to 40 days post-printing). Moreover, functional analysis revealed properties typical of immature neuronal networks.

In conclusion, the recent technology of 3D bioprinting still faces numerous challenges: on the one hand, the development of ink formulations and materials that guarantee the reproduction of complex tissue composition with specific mechanical and physical properties (such as porosity, or the coexistence of organic and inorganic building blocks necessary to reproduce the bone structures, see Refs. [73,107,108,109]). On the other hand, there is an increasing need for sophisticated printing technologies to process the needed material to model human tissues correctly. In the case of neuronal 3D structures [110,111], this technology is additionally challenged by the physiological properties of neurons, such as building a three-dimensional, often patterned network. Overall, 3D bioprinting represents a promising technology to model organized neuronal structures reproducibly. Finally, recently, 3D printing has also been employed to standardize the production of organoids [111,112,113,114,115]. Despite still being at the beginning stages, and no attempts having yet been made with neural tissues, the convergence between 3D bioprinting technology and organoids is an appealing strategy in brain research.

Organ-on-a-chip is a transformative approach that harnesses the versatility of microfluidic platforms for the engineering of new functional tissues. Spanning from this, organoids-on-a-chip are currently attracting attention, due to the optimal results obtained from the synergistic combination between organoids and microfluidic technology. Comprehensive reviews have recently detailed the latest results in brain organoid-on-chip technology elsewhere [116,117]. Among the most relevant advancements in brain organoid-on-a-chip research, the work from Karzbrun and colleagues has revealed the underlying mechanisms of folding involved in neurodevelopmental disorders [118]. The authors reported the formation of surface wrinkles during the in vitro self-organization of human brain organoids pre-assembled in microfabricated compartments. The microchip platform was engineered to host and facilitate brain organoid assembly and monitoring with in situ imaging technology during weeks of development. The folding wavelength was found to scale linearly with the thickness of the forming tissue, indicating a balance between energies associated with bending and stretching. Moving beyond the study of brain development, Wang and co-workers [119] reported the use of a novel brain organoid-on-a-chip approach to study prenatal nicotine exposure. A multi-inlet device housing a number of brain organoids was engineered to offer the possibility to screen for a single drug on multiple organoids at the same time. The overall development of the brain organoids was guided and ultimately satisfactory, resulting in the ability to recapitulate defined neural differentiation and regionalization. Nicotine exposure was found to induce premature and abnormal differentiation with the overexpression of TUJ1 and the disruption of neurodevelopment with the expression of PAX2, PAX6, FOXG1, and KROX20, among other markers indicating pathological alteration. Brain organoid-on-a-chip platforms have demonstrated unparalleled potential in generating functional models for the screening of new therapeutics. Nevertheless, the lack of available microfabrication facilities, the difficulties associated with engineering methodologies to replicate microfluidic architecture, as well as the limited number of organoids that can be cultivated at the same time are some of the pivotal factors that are currently limiting incremental research associated with brain organoid-on-a-chip technology. 

## 6. The Versatility of 3D Brain Organoids: Modeling Neurodevelopmental and Neurodegenerative Disorders Unraveling Pathophysiological Mechanisms

As we reported so far, 3D brain organoids have emerged as a potent tool for modeling human neurological disorders, including neurodevelopmental and neurodegenerative disorders, thus offering a physiologically relevant and personalized approach over 2D cell cultures [120,121]. Organoids can be derived from patient-specific iPSCs offering an opportunity to model diseases using cells from affected individuals. Moreover, organoids can recapitulate aspects of organ development, allowing researchers to study disease processes during embryonic development. In addition, organoids can be generated in large numbers, facilitating the high-throughput screening of drugs and therapeutic compounds, potentially accelerating the drug discovery process (Figure 4).

Patient-derived stem cells can be utilized to generate brain organoids of a specific pathology, considering not only the disease but also the genetic background of the patient. These advancements in organoid technology open new horizons to better decipher mechanisms involved in brain development and maturation, as well as pathologies affecting neurons and glial cells. They hold value in drug development and screening and the exploration of personalized therapies [122,123,124]. It has to be noticed that the absence of microglia and proper vascularization, including the presence of the blood–brain barrier (BBB), has been recognized as a significant limitation in brain organoids when it comes to modeling brain functions in pathological conditions, particularly in drug testing experiments [125]. The role of microglia and the immune system cannot be overlooked, as they play crucial roles in various neurological conditions. Additionally, vascularization is essential for drug distribution within the organoid, and disruptions in BBB integrity are commonly observed in neurodegenerative disorders, brain cancers, and traumatic brain injuries. These limitations underscore the need for further advancements in brain organoid technology to mimic the complex cellular interactions and pathological features of the brain more accurately [125].

Brain organoids have emerged as a valuable tool for modeling neurodevelopmental genetic disorders, allowing researchers to recreate pathologies, such as Tourette’s Syndrome [126], microcephaly, and macrocephaly, and investigate the impact of pathogens on brain development [127,128,129]. For instance, studies using induced pluripotent stem cells (iPSCs) from microcephaly patients have demonstrated smaller organoids and premature differentiation in the neural progenitor regions [128], while the deletion of the tumor-suppressor gene PTEN in human iPSCs results in the formation of abnormally large organoids with the over-proliferation and delayed neurogenesis of neural progenitor cells [129]. Brain organoids have also proved to be valuable for investigating viral infections, including the Zika virus, the herpes simplex virus (HSV), and the cytomegalovirus. In brain organoids, the Zika virus specifically targets SOX2-positive neural progenitor cells, leading to suppressed proliferation, increased cell death, and a significant reduction in organoid size [17,129,130]. Similarly, early-stage brain organoids composed of human iPSC-derived neural rosettes infected with HSV-1 exhibit a loss of structural integrity and neuronal alterations [131]. These studies highlight the utility of brain organoids in studying the effects of viral infections on brain development and function [131,132,133,134,135,136].

Brain organoids have become a valuable tool for studying neurodevelopmental disorders such as schizophrenia (SZ), and fragile X syndrome (FXS). SZ is a complex and severe neuropsychiatric disorder associated with a wide range of debilitating symptoms. Many aspects of its multifactorial complexity are still unknown, and some are accepted to be an early developmental deficiency with a more specifically neurodevelopmental origin. The neurodevelopmental hypothesis suggests that the interactions of multiple genes trigger a cascade of neuropathological events during the embryonic and post-natal development of the brain that may be initiated by environmental factors such as maternal infections or infectious agents associated with the onset of inflammatory responses [137] that trigger symptoms in early adolescence [138] and lead to the emergence of psychosis at the time of the transition from late adolescence to young adulthood. The disorder can be heritable and polygenic, with risk alleles distributed widely across the genome [139] and people with SZ are enriched by rare copy variants (CVs) in genes associated with neurodevelopmental disorders, particularly autism spectrum disorders and intellectual disability [140]. CVs can disrupt gene function by increasing or decreasing gene dosage, and one recurrent CV is a deletion in the 22q11.2 region that typically encompasses around 50 protein-coding genes and is the most common CV in humans [141]. This deletion causes the 22q11.2 deletion syndrome (DS), a neurodevelopmental disorder that is one of the most frequent genetic risk factors for SZ [142,143], and, like SZ, the 22q11.2 DS patients display neurodevelopmental delays and cognitive dysfunctions [144]. One challenge facing researchers trying to elucidate the mechanisms underlying SZ and 22q11.2 DS is that the molecular and cellular processes governing embryonic and early postnatal brain maturation are still unclear.

Recent studies on mice models of 22q11.2 DS have provided important insights [143,145,146,147] including a description of circuit dysfunctions [148,149,150] and the implication of mitochondrial dysfunctions in the onset of cognitive phenotypes [151,152,153]. However, the molecular and cellular mechanisms leading to human neuronal phenotypes remain poorly understood. In recent years, some studies have begun to explore transcriptional changes and the implication of mitochondrial dysfunctions in neural cells derived from 22q11DS patients [154,155,156], but the functional defects in human 22q11.2 DS neurons and the underlying mechanisms have not been investigated. Therefore, understanding the timepoints of neuronal and, eventually, glial cell phenotypes during cell differentiation and maturation processes could lead to an insight into the development of the cellular phenotypes associated with the disorder.

Cerebral organoids provide a promising tool to investigate the molecular and cellular alterations involved in the onset of 22q11.2 DS, exploiting the possibility to directly use patient-derived IPSCs which consider not only the footprint of the disease but also the genetic background of the patient, enabling the evaluation of specific early pathological targets for drug development. Khan and colleagues successfully generated 3D cerebral cortical organoids from 22q11.2 DS-derived IPSCs. They analyzed the transcriptional profile along 100 days of differentiation and found an altered expression of several genes, related to RNA modification and RNA silencing, mitochondria functions, and neuronal excitability, including those coding for calcium transport. Electrophysiological and calcium imaging experiments confirmed changes in spontaneous firing and depolarization-induced calcium signaling in 22q11.2 DS cortical neurons due to abnormalities in the resting membrane potential and defects in the function of the voltage-gated L-type calcium channels. Interestingly, the voltage-gated L-type calcium channels have been identified as common risk genes for SZ [157], thus suggesting that alterations of cytoplasmic calcium signaling may be one of the key targets for drug development.

FXS, the most common single-gene cause of autism spectrum disorder (ASD), is characterized by an expansion of the CGG triplet in the FMR1 gene, leading to the loss of the FMR1 protein [158]). The FMRP protein, which is involved in RNA regulation, plays a crucial role in neuronal development, synaptic plasticity, and dendritic spine architecture [159], and recent studies on FXS-patient-derived neural progenitor cells (NPCs) have indeed revealed abnormal gene expressions related to protein synthesis, neural development, and migration [160,161,162,163]. Differential gene expression analysis revealed 218 differentially expressed genes involved in cell fate commitment and differentiation, with down-regulated genes related to fate specification, migration, differentiation, and maturation, and up-regulated genes associated with cell proliferation [164]. A recent study by Kang et al. [165] observed alterations in neural proliferation and differentiation in brain organoids derived from FXS patients’ iPSCs during a developmental period corresponding to mid-fetal human brain development. In FXS, alterations in gene expressions related to synapse development, axon targeting, and cytoskeleton organization [166] result in an imbalance between excitatory and inhibitory signaling [28,167,168,169,170,171].

FMRP knockout brain organoids have shown significant phenotypes of immature astroglial cells by revealing an increased number of GFAP-positive cells [28], a situation similar to the FXS postmortem brain tissues [172,173]. Interestingly, the gene of FMR1 is also part of group I of metabotropic glutamate receptor (mGlu) signaling, and FMR1 together with the mGlu type 5 and the scaffold proteins Homer1b and Shank3 are highly expressed in immature astrocytes [174,175], but their expression decreases over the differentiation and maturation of the cells, thus suggesting that they can play crucial roles during the acquisition of the astrocytic phenotype [176]. Indeed, accumulating evidence indicates that mGluR5 signaling in immature astrocytes controls astrocyte morphogenesis [177], thus suggesting that abnormal astrocyte maturation may also occur in the absence of FMR1 in the brain organoids [28]. These findings highlight the potential of brain organoids in unraveling the underlying mechanisms of neurodevelopmental disorders, particularly providing insights into the pathophysiology of FXS.

The use of “fused” dorsal–ventral forebrain organoids, also known as assembloids, generated from iPSCs, has provided valuable insights into the cellular and molecular mechanisms underlying Timothy syndrome (TS) [54,178]. TS is a rare and severe neurodevelopmental disorder caused by a mutation in an L-type calcium channel subunit. Calcium influx through voltage-gated calcium channels is involved in regulating processes such as cytoskeletal dynamics, cell adhesion, and signaling pathways crucial for neuronal migration [179], and in TS-patient-derived brain organoids, Birey et al. [180] showed the inefficient migration of cortical interneurons and increased calcium signaling following depolarization. These findings shed light on the role of calcium signaling in interneuron migration and the establishment of neural circuits. To gain a more comprehensive understanding of TS pathophysiology, Revah et al. [181] transplanted TS brain organoids into the cerebral cortex of newborn athymic rats. This transplantation approach enabled the investigation of organoid maturation and integration with a functional neural circuit, leading to the manifestation of disease-related phenotypes. These models offer valuable insights into the pathophysiology of TS and have the potential to identify novel therapeutic targets for this neurodevelopmental disorder.

Another promising tool for the study of Tourette’s syndrome is represented by basal ganglia organoids [182]. Tourette’s syndrome is a neuropsychiatric disorder characterized by uncontrollable motor or vocal tics that manifest in childhood and that very often co-occur with obsessive-compulsive disorder (OCD). Tourette’s syndrome and OCD are compulsive repetitive behaviors that are considered to be neurodevelopmental in origin, but their etiology and pathophysiology are unknown. Convergent neuroimaging and neurophysiological findings support a model in which abnormal cortico-striato-thalamo-cortical (CSTC) loops [183,184] and abnormalities in the basal ganglia may play a crucial role in the onset of symptoms [185]. Supporting the CSTC loop hypothesis, it is known that the optogenetic activation of the prefrontal cortex (PFC) induces excessive grooming in rodents [184] and that dopamine, one important modulator in CSTC circuits, plays a role in the onset of OCD symptoms in patients [186,187] and of stereotypical behavior in rodents [188,189]. There is also strong evidence that the dopaminergic system plays a key role in Tourette’s syndrome [190] and there might be some therapeutic role for dopamine D2 receptor blockers in OCD [191]. The current CSTC model of OC-like behavior and the hypothesis of dopaminergic involvement are both entirely neurocentric, but recent studies have identified a role for astrocytes in the modulation of dopamine homeostasis in the PFC [176] and revealed the importance of these cells in a number of behaviors, including repetitive behaviors [189,192], thus indicating that animal behavior is not a result of neuronal activity alone but requires the coordinated activity of neurons and astrocytes. Overall, these studies on rodent models of OC-like disorders have provided important insights, but human neuronal and astroglial phenotypes remain poorly described.

To better understand the neuronal pathways leading to OC-like disorders, recent studies have explored the early developmental pathophysiology of Tourette’s syndrome by using iPSC-derived basal ganglia organoids [182]. Repetitive behaviors and tic release elicit prominent activity in the basal ganglia [193], and neuroimaging data suggest decreases in the striatal volume in Tourette’s syndrome patients [193]. The basal ganglia organoids exhibit impaired development of medial ganglionic eminence and reduced differentiation of cholinergic and GABAergic interneurons. The transcriptome analysis revealed the mis-patterning of the ventral telencephalon with a relative lack of ventromedial progenitors accompanied by enhanced dorsolateral fates. This results in the developmental loss of interneurons, suggesting that the interneuron loss noted in the postmortem basal ganglia of patients with Tourette’s syndrome is a potential consequence of an inherent tendency of the basal ganglia to undergo different regional specifications. These findings contribute to our understanding of Tourette’s syndrome etiology and offer a developmental lens to the pathologies currently associated with Tourette’s syndrome such as OC-like disorders.

Brain organoids offer researchers a valuable platform to study the molecular mechanisms underlying neurodegenerative diseases such as Alzheimer’s (AD) and Parkinson’s (PD), enabling the observation of changes in gene expression, protein function, and cellular interactions. It is noteworthy that neurodegenerative diseases are commonly associated with late-onset symptoms and are linked to the aging process. However, emerging evidence suggests that pathological mechanisms, such as amyloid-β accumulation in AD or changes in brain structure and function in PD, can initiate decades before symptom onset [194,195,196,197]. Moreover, genetic predisposition can lead to the development of neuronal and glial dysfunctions at an earlier age. By utilizing organoid technology, researchers have successfully recapitulated neuropathological hallmarks associated with AD, including amyloid-β deposition, the hyperphosphorylation of tau protein, continuous aggregation in 3D models, neuroinflammation, and gliosis [25,194,198,199]. Most of the contributions come from brain organoids modeling familial AD, which represents a small fraction (<5%) of AD cases caused by genetic variants in the amyloid precursor protein (APP) gene or presenilin genes (PSEN1, PSEN2), while the majority of cases (sporadic AD) result from a combination of multiple factors [200,201]. These organoids can also be gene-edited to express specific genetic variations or those derived from patient-derived iPSCs. Abnormalities in tau protein, synaptic dysfunction, disturbance in the balance of excitatory and inhibitory neuronal circuits, and the presence of amyloid-β plaques contribute to enhanced neuronal hyperactivity observed in AD organoids compared to wild type [202,203]. Co-cultures of microglia and brain organoids have been utilized to study the interaction between microglia, neural cells, and pathological features of Alzheimer’s disease. Specifically, the role of APOE4, the strongest genetic risk factor for late-onset sporadic AD, has been investigated in co-culture models, revealing a reduced Aβ plaque clearance ability compared to APOE3 [204]. AD brain organoids have also been employed as a drug platform, demonstrating a reduction in amyloid-β aggregates through the use of β- or γ-secretase modulators, along with mitigating the effect of tau pathology [194].

Human-derived 3D models of Parkinson’s disease (PD) offer a close resemblance to the disease phenotype, exhibiting characteristic features such as the shortened neurite length of dopaminergic neurons and α-synuclein aggregation, surpassing the limitations of 2D and animal models [205]. Brain organoids derived from PD patients carrying missense mutations in the LRRK2 gene, associated with late-onset disease, demonstrate impaired dopaminergic neuron development and reduced astrocytic activity [206,207]. Additionally, brain organoids have been employed to investigate the molecular mechanisms underlying α-synuclein deposition, revealing a correlation with LRRK2 mutation [205]. Furthermore, PD organoids have been instrumental in studying the juvenile form of PD by introducing mutations into the DNAJ6 gene. Mutant organoids exhibit impaired neurodevelopment, reduced dopamine release, increased oxidative impairment, and dysfunctions in mitochondria and lysosomes [208].

Over the past decade, a number of studies have highlighted the crucial roles of inflammatory processes in many brain diseases, highlighting a possible role of reactive glial cells in the pathophysiology of neurodegenerative disorders. Astrocytes are indeed involved in all forms of brain disease and lesions, to which they respond by undergoing a series of cellular, molecular, and functional changes known as “reactive astrogliosis” [209]. These alterations may be transient or long-lasting, and interdisciplinary approaches combining omics with physiology and genetic manipulations have shown that they can have both harmful and beneficial effects, thus indicating that astrocytes have multiple states of reactivity and functions depending on the myriad of intrinsic and extrinsic cues governing their post-injury gene expression and function [209]. Many aspects of astrocyte functioning have been unveiled from studies conducted in murine models; however, growing evidence shows many differences between mouse and human astrocytes starting from their development and encompassing morphological, transcriptomic, and physiological variations when they achieve complete maturation. The study of human reactive astrocytes has, therefore, been limited by the availability of resources and, more importantly, because there are important transcriptional and functional differences between rodent and human astrocytes [127,210]. In a recent study, Cvetkovic et al. address this technical shortcoming by developing bio-engineered neural organoid cultures containing mature astrocytes which allow for the investigation of the dynamics of astrocyte reactivity and its downstream effects on neuronal activity [211,212]. The authors successfully generated multicellular organoid systems containing astrocytes that exhibited key features of mature cells. After extensively validating the morphology and gene expression of astrocytes, Cvetkovic et al. investigated the effects of over-activating calcium signaling in astrocytes on neuronal activity. They successfully found that the chronic activation of astrocytes resulted in changes in gene expression like those observed in reactive astrocytes, suggesting aberrant calcium signaling in inducing a reactive phenotype. However, the observed reactivity did not display neurotoxicity as seen in other studies, possibly due to differences in experimental conditions or the requirement of additional disease-associated stressors [211]. Overall, this new in vitro model is promising for studying the dynamics of human reactive astrocytes and their impact on neuronal function [212]. Future research should explore more complex models that incorporate other cell types involved in neuroinflammatory processes to better understand the contribution of reactive astrocytes to disease progression and test potential therapies [212].

## 7. Conclusions

Over the last decade, 3D brain cultures, such as brain organoids, have emerged as powerful tools for studying neural function and modeling neurological diseases. Brain organoids are self-organized 3D structures that recapitulate key aspects of human brain development and pathology [5]. One of the key advantages of brain organoids is that they can be generated from patient-specific pluripotent stem cells, providing a personalized approach to studying disease mechanisms and developing new therapies [213].

There are several methods for generating brain organoids, including self-assembly, guided differentiation, and bioprinting. Self-assembled organoids are formed by allowing cells to self-organize into 3D structures without external manipulation. In contrast, guided differentiation involves the use of growth factors and other signaling molecules to drive cell differentiation and tissue patterning. Bioprinting allows precise control over the spatial arrangement of cells and extracellular matrix components, enabling the generation of complex, multi-layered structures [17].

While each method has its advantages and limitations, the choice of method should be tailored to the specific research question. For example, self-assembled organoids may better recapitulate the complex interactions between different cell types and signaling pathways that occur during brain development, whereas bioprinted organoids may provide more precise control over tissue structure and composition [62].

Incorporating microglia and vascularization into brain organoids can further enhance their relevance as disease models and improve drug development outcomes. Microglia are the resident immune cells of the central nervous system, and they play a critical role in neuroinflammation and neurodegeneration. Recent studies have shown that microglia can innately develop within cerebral organoids, providing a new platform for studying their function in the context of neurodegenerative diseases [62]. In addition, incorporating a vascular network into brain organoids can improve nutrient and oxygen delivery, allowing for longer-term culture and more accurate modeling of disease progression and drug response [213].

We also need to consider some ethical issues surrounding advancements in brain organoid research, particularly as it moves towards creating more complex models of mature human cortical regions and their interconnectedness. One significant concern revolves around the potential emergence of consciousness in brain organoids or the complex brain assembloids that combine organoids from multiple cell lineages [214].

While these concerns exist, several factors mitigate them. It is difficult to compare organoid brain waves with those of developing human brains. The neural correlates of consciousness are complex and involve diverse brain regions. The term “consciousness” has different meanings, and the ethical implications depend on the specific definition. Organoids lack the necessary structures for complex consciousness. The moral concerns may be influenced by the human origin of organoids. However, the absence of social interaction and language acquisition makes conscious self-awareness unlikely. Thus, the ethical challenges related to consciousness in organoids are not significant in current research.

In conclusion, 3D brain cultures offer a promising platform for studying neural function and developing new therapeutic strategies for neurodevelopmental and neurodegenerative disorders. Self-assembled, guided, and bioprinted organoids each have unique advantages and limitations, and the choice of method should be tailored to the specific research question. Incorporating microglia and vascularization into brain organoids can enhance their relevance as disease models and improve drug development outcomes. As the field continues to advance, it is likely that 3D brain cultures will play an increasingly important role in unlocking the secrets of neural function and developing effective treatments for neurological disorders.

## Figures and Tables

**Figure 1 ijms-24-10762-f001:**
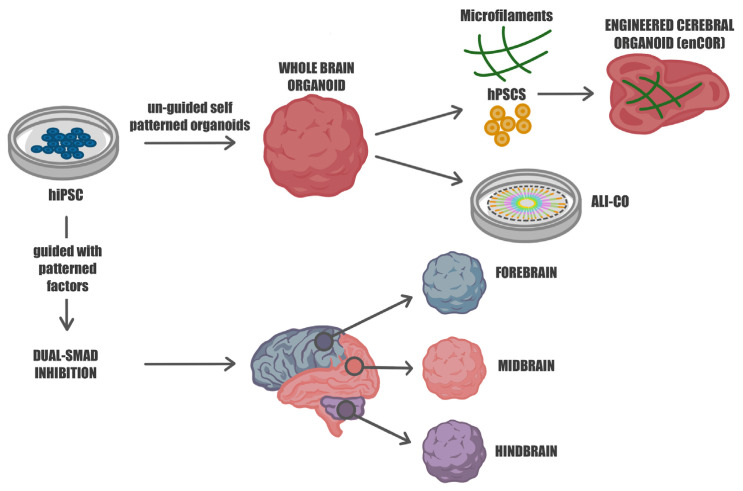
Guided and unguided approaches for differentiating human iPSCs into diverse neural organoids. Human hiPSCs can be cultivated in self-organizing 3D cultures to yield either unguided neural organoids, also known as whole-brain organoids, or regionalized neural organoids through guided approaches. The unguided methods (**top**) rely on the intrinsic signaling and self-organization capacities of hiPSCs, leading to their spontaneous differentiation into organoids that closely resemble developing brain tissues. These organoids exhibit heterogeneity, representing various brain regions. To mitigate this variability and enhance cell survival, microfilament-engineered cerebral organoids (enCORs) and air–liquid interface cerebral organoids (ALI-COs) have been developed. The guided approaches (**bottom**) utilize specific small molecules and growth factors to generate spheroids that predominantly represent a particular tissue type, reducing heterogeneity and promoting standardization.

**Figure 2 ijms-24-10762-f002:**
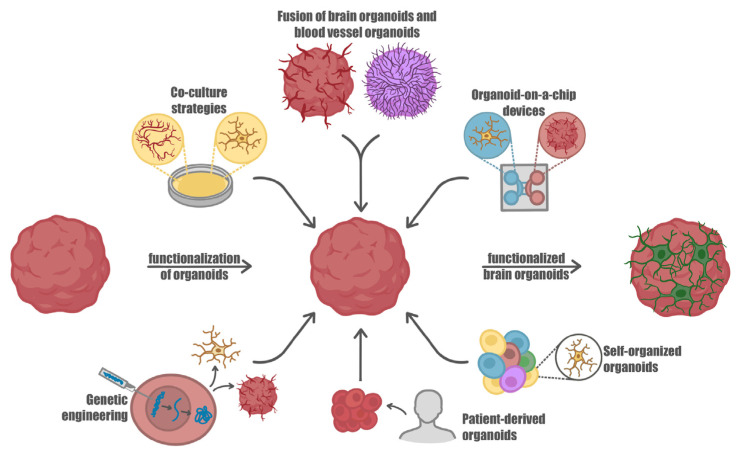
Advancement in brain organoids functionalization. IPS-derived brain organoids represent valuable in vitro models to study relevant physiological mechanisms underlying brain development and functions as well as to investigate pathologies which affect the nervous system. Advances in culturing and differentiating induced pluripotent stem cells (iPSCs) led to the possibility of integrating brain organoids with different cell types, including those that do not originate from the neuroectodermal line, such as microglia, providing a valuable platform to investigate the neuro-immune crosstalk. Moreover, the integration of a vascular system within the organoid structures improves nutrient delivery, waste removal, and oxygenation, closely mimicking the physiological conditions found in real tissues. In addition, patient-derived stem cells have been used to generate brain organoids with a specific pathology, thus paving the way for personalized therapies. These advancements in organoid technology open new horizons to better decipher processes involved in brain development and maturation, as well as pathologies affecting neurons and glial cells.

**Figure 3 ijms-24-10762-f003:**
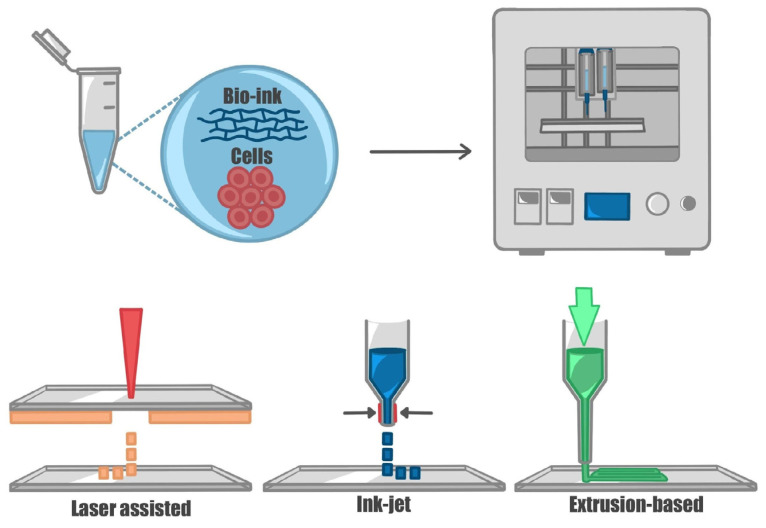
Three-dimensional bioprinting techniques. The figure represents the main printing techniques used for printing cells. Cells are encapsulated in a bio-compatible material (bio-ink) and printed. Through laser-assisted printing, the material is transferred from a donor to a receiver slide through a laser; in ink-jet printers, the material is dispensed drop-wise applying a force (thermal, acoustic, or mechanical) at the dispenser tip; extrusion-based techniques allow for the continuous extrusion of the material through the nozzle.

**Figure 4 ijms-24-10762-f004:**
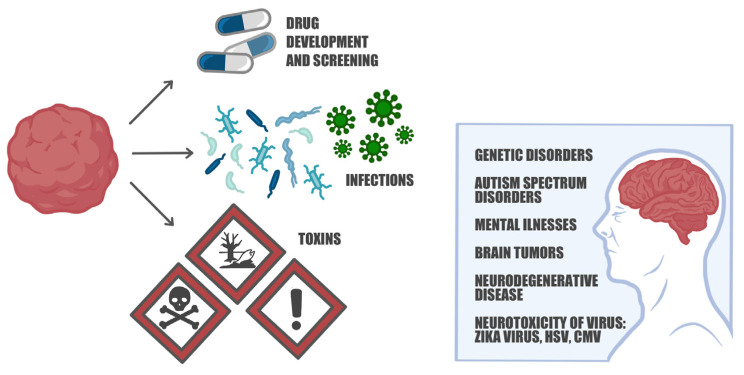
Brain organoids: modeling brain disorders, drug screening, and immune response investigations. Brain organoids serve as sophisticated models that replicate the intricate cellular interactions and developmental processes of the brain. They have proven invaluable in unraveling the underlying mechanisms of various conditions such as Fragile X syndrome, 22q11.2 deletion syndrome, Tourette’s syndrome, Alzheimer’s disease, and Parkinson’s disease. Moreover, these organoids provide a valuable platform for screening potential drug candidates and identifying therapeutic agents. Researchers can also infect organoids with viruses or expose them to toxins to study the impact on neural cells and investigate the subsequent immune response. These capabilities make brain organoids a versatile tool for advancing our understanding of brain disorders and exploring novel treatment strategies.

**Table 1 ijms-24-10762-t001:** Comparison of neurospheres and brain organoids in 3D neural cell culture.

Pros of Neurospheres	Pros of Brain Organoids
- Three-dimensional aggregates of differentiated neural cells or neural progenitor cells derived from isolated primary tissue or iPSCs	- Organized 3D structures derived from embryonic stem cells or induced pluripotent stem cells (iPSC)
- Homogeneous population of neural cells	- Heterogeneous population of neural cells that closely resemble the composition of the developing brain
- Simple and rudimentary structures	- Complex structures
- Rapid growth for a limited period of time	- Maturation over an extended period of time
- Convenient system for studying early neural development and basic cellular and molecular processes (e.g., effects on gene expression, signaling pathways, drug treatment, proliferation, and differentiation potential)	- Platform for more sophisticated analysis (e.g., electrophysiological activity, synaptic connections, disease modeling, drug screening, and testing)

**Table 2 ijms-24-10762-t002:** Comparison of hydrogels in bioprinting neural tissue. Table comparing different hydrogel precursors and bioink compositions used for the printing of neural tissues. In addition to laser-assisted, inkjet, and extrusion-based techniques, other types of printing techniques cited in this table include: microvalve bioprinting, a drop-on-demand technique based on a robotic platform capable of moving along three axes, accompanied by an array of multiple electromechanical microvalve print-heads; stereolithography bioprinting, a technology that utilizes light to precisely create three-dimensional structures by layering and polymerizing light-sensitive materials.

Cells	Hydrogel Precursor	Bioprinting Technology	Features	Ref.
Human neural stem cells	Agarose Alginate Carboxymethyl chitosan	Extrusion	In vitro poor stability Spontaneously active neurons	[75]
iPSC derived NPC	Fibrin Chitosan Alginate	Microfluidic-assisted extrusion	Neuronal maturation with microsphere-released morphogens	[77]
Primary rat cortical neurons	Gellan-gum RGD	Extrusion	Stability in vitro Dendrite extension Ability to layer neurons in hierarchical constructs	[88]
Rat embryonic neurons and astrocytes	Type I collagen	Microvalve	Multilayered scaffold Neurite extension	[89]
Neural stem cells	GelMA + graphene	Stereolithography	Limited promotion of neuronal differentiation	[90]
sNPC and OPC	Matrigel Gelatin/fibrin blend GelMA PEGDA + AG/MC	Extrusion	Axon propagation NPC maturation No OPC maturation or axon myelination Neuronal spontaneous and induced activity	[85]
iPSC-derived cortical neurons and glial cells.	Matrigel/Alginate	Extrusion	Cell viability up to 70 days post-printing Functional neural network	[92]

iPSC = induced pluripotent stem cell; NPC = neural precursor cell; RGD = arginine-glycine-aspartic acid; GelMA = gelatin methacrylamide; sNPC = spinal NPC; OPC = oligodendrocyte progenitor cell; PEGDA = poly(ethylene glycol) diacrylate; AG/MC = alginate mixed with methylcellulose.

## Data Availability

Not applicable.

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
