# Peer review of "Unlocking Neural Function with 3D In Vitro Models: A Technical Review of Self-Assembled, Guided, and Bioprinted Brain Organoids and Their Applications in the Study of Neurodevelopmental and Neurodegenerative Disorders"

_ijms, 2023, doi:10.3390/ijms241310762_

Round 1
Reviewer 1 Report
The review provides a complete and detailed description of the use of organoids for the study of the brain and to address important questions for the research in neurodevelopmental and neurodegenerative disorders. The authors start describing the discovery of the IPSCs and how from these preparation derived the genesis of organoids, explaining the steps for the generation of the 3D structure in vitro for different brain structures. They also describe the limits of these structures and how new methods have been developed to overcome these limits. The authors match the historical perspective to the technical developments, providing details on the specific molecules used in the different steps and on the different brain cells that can be found in the organoids. Finally, they report in detail a limit recently overcome, which is the possibility of inserting microglia and vascularized structures in these preparations.
Then they describe the use of organoids in bioprinting and its potential application in the treatment of injuries of the nervous system. The bioprinting is a link between the basic description of organoids and their use in modelling brain disorder. The authors spend the second part of the review describing different models that have been generated for the study of: autism spectrum disorders, obsessive compulsive disorder, Alzheimer disease and the study of inflammatory disorders.
Overall, this is a well done and complete review that properly describes the use of organoids for the study of brain function in health and disease.
The only minor change that I would suggest is to add in the conclusion paragraph a comment on the ethical implication of this discovery, since the continuous advances in the technique are improving the similarities with the human brain.
Author Response
REBUTTAL LETTER
We deeply thank the editors and reviewers for their careful evaluation of our work and for prompting us to improve the quality of the manuscript.
To take into account the reviewer's suggestions, we have modified the text and the reference list in response to specific questions raised by the reviewers. We also added two Tables.
Dr. Gianluca Cidonio, who contributed to the revision, and specifically wrote the Organ on Chip paragraph and the bioink table, has been included in the authors’ list.
Below, you will find the point-by-point response to each comment.
We believe that this text improvement, as per the reviewer's suggestions, will make our MS suitable for publication in for publication in the SI "Recent Advance in 3D Cultures” of IJMS.
Reviewer #1
The review provides a complete and detailed description of the use of organoids for the study of the brain and to address important questions for the research in neurodevelopmental and neurodegenerative disorders. The authors start describing the discovery of the IPSCs and how from these preparation derived the genesis of organoids, explaining the steps for the generation of the 3D structure in vitro for different brain structures. They also describe the limits of these structures and how new methods have been developed to overcome these limits. The authors match the historical perspective to the technical developments, providing details on the specific molecules used in the different steps and on the different brain cells that can be found in the organoids. Finally, they report in detail a limit recently overcome, which is the possibility of inserting microglia and vascularized structures in these preparations.
Then they describe the use of organoids in bioprinting and its potential application in the treatment of injuries of the nervous system. The bioprinting is a link between the basic description of organoids and their use in modelling brain disorder. The authors spend the second part of the review describing different models that have been generated for the study of: autism spectrum disorders, obsessive compulsive disorder, Alzheimer disease and the study of inflammatory disorders.
Overall, this is a well done and complete review that properly describes the use of organoids for the study of brain function in health and disease.
The only minor change that I would suggest is to add in the conclusion paragraph a comment on the ethical implication of this discovery, since the continuous advances in the technique are improving the similarities with the human brain.
We thank the reviewer for their valuable feedback on our review. We appreciate their positive assessment of the paper and are glad to hear that they found it to be a comprehensive and detailed description of the use of organoids in studying the brain and addressing important questions related to neurodevelopmental and neurodegenerative disorders. We have carefully considered their suggestion to include a comment on the ethical implications of this discovery in the conclusion paragraph, and we agree that it is an important aspect to address. Therefore, we have revised the conclusion to incorporate this point. (Lines 908-921)
Reviewer 2 Report
This review with the title is a very well written summary of the current knowledge in the field of 3D brain models including organoids and bioprinting. I only have minor editing suggestions:
1) It would be great to add an abstract about the application of Neurospheres.
2) To further emphasize the pros and cons of organoids vs spheres it would be great to visualize it with a table (e.g use of cell lines vs primary cells etc.)
3) “Hydrogels that have been used to print neural cells include alginate, agarose, chitosan ([66]), gellan gum-RGD [77], collagen [78], modified gelatin GelMa [79], and Matrigel [75], [80]. “
It would be great if you could ellaborate more on the specific citations. What worked best? How long were the neurons viable in these distinct hydrogels? What extrusion method was used?Please make a table comparing different hydrogels in bioprinting and their successes (viability, long-term survival, self-assembly? etc.)
4) Please add the following paper as it was an important contribution to the field:
"Amplification of human interneuron progenitors promotes brain tumors and neurological defects" Science 2022
5) The abstract concerning brain on a chip devices is a bit scarce. Maybe you want to ellaborate a bit one.
Author Response
REBUTTAL LETTER
We deeply thank the editors and reviewers for their careful evaluation of our work and for prompting us to improve the quality of the manuscript.
To take into account the reviewer's suggestions, we have modified the text and the reference list in response to specific questions raised by the reviewers. We also added two Tables.
Dr. Gianluca Cidonio, who contributed to the revision, and specifically wrote the Organ on Chip paragraph and the bioink table, has been included in the authors’ list.
Below, you will find the point-by-point response to each comment.
We believe that this text improvement, as per the reviewer's suggestions, will make our MS suitable for publication in for publication in the SI "Recent Advance in 3D Cultures” of IJMS.
Reviewer #2
This review with the title is a very well written summary of the current knowledge in the field of 3D brain models including organoids and bioprinting. I only have minor editing suggestions:
- It would be great to add an abstract about the application of Neurospheres.
We thank the reviewer for their valuable feedback on our review. Following the reviewer’s suggestion we added a paragraph on the development of neurospheres. (Lines 103-146)
- To further emphasize the pros and cons of organoids vs spheres it would be great to visualize it with a table (e.g use of cell lines vs primary cells etc.)
Following the reviewer’s suggestion we added a scheme summarizing the advantages of organoids and spheres. (new Table 1)
3) “Hydrogels that have been used to print neural cells include alginate, agarose, chitosan ([66]), gellan gum-RGD [77], collagen [78], modified gelatin GelMa [79], and Matrigel [75], [80]. “
It would be great if you could ellaborate more on the specific citations. What worked best? How long were the neurons viable in these distinct hydrogels? What extrusion method was used?Please make a table comparing different hydrogels in bioprinting and their successes (viability, long-term survival, self-assembly? etc.)
Following the reviewer’s suggestion, we have now improved the indicated section introducing a comprehensive table comparing the use of hydrogels specifically for the printing of neural tissue. (Table 2)
4) Please add the following paper as it was an important contribution to the field:
"Amplification of human interneuron progenitors promotes brain tumors and neurological defects" Science 2022
Following the reviewer’s suggestion we added the reference.
5) The abstract concerning brain on a chip devices is a bit scarce. Maybe you want to elaborate a bit one.
We thank the reviewer for this comment. We have now implemented a full paragraph on brain organoid-on-a-chip expanding in the biofabrication section. As the organoid-on-a-chip technology as been extensively reported in recently published reviewes (1. Castiglione, H.; Vigneron, P.-A.; Baquerre, C.; Yates, F.; Rontard, J.; Honegger, T. Human Brain Organoids-on-Chip: Advances, Challenges, and Perspectives for Preclinical Applications. Pharmaceutics 2022, 14, 2301. https://doi.org/10.3390/pharmaceutics14112301. 2. Jiyoung Song, Seokyoung Bang, Nakwon Choi, Hong Nam Kim; Brain organoid-on-a-chip: A next-generation human brain avatar for recapitulating human brain physiology and pathology. Biomicrofluidics 1 December 2022; 16 (6): 061301. https://doi.org/10.1063/5.0121476) we have included the most relevant studies in the last 5 years to provide an overview of the microfluidic technology to the reader. (Lines 589-617).